# Wearable Piezoelectric-Based System for Continuous Beat-to-Beat Blood Pressure Measurement

**DOI:** 10.3390/s20030851

**Published:** 2020-02-05

**Authors:** Ting-Wei Wang, Shien-Fong Lin

**Affiliations:** 1Department of Electrical and Computer Engineering, College of Electrical and Computer Engineering, National Chiao Tung University, Hsinchu 30010, Taiwan; w756704@gmail.com; 2Institute of Biomedical Engineering, College of Electrical and Computer Engineering, National Chiao Tung University, Hsinchu 30010, Taiwan; 3Krannert Institute of Cardiology, Indiana University School of Medicine, Indianapolis, IN 46202, USA

**Keywords:** continuous blood pressure, piezoelectric sensor, wearable device

## Abstract

Non-invasive continuous blood pressure measurement is an emerging issue that potentially can be applied to cardiovascular disease monitoring and prediction. Recently, many groups have proposed the pulse transition time (PTT) method to estimate blood pressure for long-term monitoring. However, the PTT-based methods for blood pressure estimation are limited by non-specific estimation models and require multiple calibrations. This study aims to develop a low-cost wearable piezoelectric-based system for continuous beat-to-beat blood pressure measurement. The pressure change in the radial artery was extracted by systolic and diastolic feature points in pressure pulse wave (PPW) and the pressure sensitivity of the sensor. The proposed system showed a reliable accuracy of systolic blood pressure (SBP) (mean absolute error (MAE) ± standard deviation (SD) 1.52 ± 0.30 mmHg) and diastolic blood pressure (DBP, MAE ± SD 1.83 ± 0.50), and its performance agreed with standard criteria of MAE within 5 mmHg and SD within ±8 mmHg. In conclusion, this study successfully developed a low-cost, high-accuracy piezoelectric-based system for continuous beat-to-beat SBP and DBP measurement without multiple calibrations and complex regression analysis. The system is potentially suitable for continuous, long-term blood pressure-monitoring applications.

## 1. Introduction

Blood pressure measurement is an essential technique used in cardiovascular disease monitoring. The auscultatory and oscillometric methods, which use the cuff sensor to detect systolic blood pressure (SBP) and diastolic blood pressure (DBP), are the golden benchmark and commonly used in clinical diagnosis. However, the cuff-based technique only provides one-shot data without continuous blood pressure information and causes inconvenience due to repeated cuff inflations. Therefore, a significant limitation of the cuff-based approach is the lack of continuous monitoring of adverse events for hypertensive patients. On the other hand, the arterial cannula is a precise method to directly measure continuous blood pressure, but the arterial cannula method uses a catheter inserted into the blood vessel that could induce potential risk by such an invasive method [1]. Consequently, the non-invasive continuous blood pressure monitoring remains a significant clinical unmet need.

Many groups have proposed methods based on pulse transition time (PTT) for continuous blood pressure monitoring [2,3,4]. PTT represents the estimated propagation time of a pulse wave measured between two sites of the artery [5,6]. PTT is frequently extracted from two-channel physiological signals, including electrocardiography (ECG) and photoplethysmography (PPG). PTT is determined by the time delay between the R-peak of the ECG signal and the time point of the maximum slope of the PPG waveform [7,8]. The PTT can also be estimated from two-channel PPG signals recorded from two different locations to calculate the time delay between both waveforms [9,10]. Many studies presented several PTT-based mathematical models for blood pressure estimation by different models [11], including logarithmic model [12], linear model [13], inverse square model [14], and inverse model [15]. The regression coefficients of these models were obtained by multiple calibrations between the oscillometric method (the gold standard) and PPT-based technique to determine the accuracy of the model. At present, the application of PTT-based methods is limited by technical considerations, including the appropriate two-channel physiological measurement, non-specific estimation models and a complex procedure of multiple calibrations.

In order to deploy a wearable device with a one-channel physiological measurement, some studies provided a pressure sensor to achieve continuous blood pressure monitoring. The pulsation changes in the artery can be detected by a pressure sensor such as the capacitive pressure sensor and piezoelectric sensor, which were usually used in blood pressure applications depending on the pressure sensitivity of the device. The capacitive pressure sensor uses the mechanism of distance change between two parallel plates to estimate blood pressure change in the artery. However, capacitive pressure sensors are typically characterized by low-pressure sensitivities due to the small stress from arterial compression. Kim et al. [16] presented a modified structure of the dielectric layer by PDMS spacer and wrinkle gold foil to improve the sensitivity of the capacitive pressure sensor for blood pressure measurement application. Compared to the capacitive pressure sensor, a piezoelectric sensor directly converts the pressure signals into electrical signals by pressure sensitivity (mV/mmHg) of the piezoelectric sensor. The radial artery of the wrist is the common site to measure continuous pressure pulse wave (PPW) signals by a piezoelectric sensor. For example, Liu et al. [17] demonstrated a PPW-based blood pressure estimation models by piezoelectric sensor implementation and a linear regression method to demonstrate the low mean absolute error (MAE) of blood pressure measurement, compared to the cuff method. Recently, Kaisti et al. [18] developed a wearable microelectromechanical system (MEMS) that produced a high correlation of mean arterial pressure (MAP) between the non-invasive pulse measurement and the invasive pulse waveform.

Most of these previous studies sought to increase accuracy through complicated mathematical models and calculations. Alternatively, an intuitive computation and reliable accuracy are preferable for practical applications in continuous beat-to-beat blood-pressure measurement. The main aim of this paper is to develop a wearable piezoelectric-based system to provide an intuitive strategy that uses the summation of initial blood pressure and pressure change for beat-to-beat SBP and DBP measurement without the cumbersome multiple calibrations.

## 2. Measurement Strategy

The measurement strategy uses the sum of the initial blood pressure measured by a cuff-based sensor (HEM-1000, OMRON) with oscillmotric method and pressure changes obtained by a calibrated piezoelectric sensor to achieve beat-to-beat SBP and DBP monitoring, according to Equation (1).
(1)$$P(t)\;=\;P_{initial}+\;\Delta P\;$$

The changes in the pressure of the artery can be calculated by the pressure sensitivity of the piezoelectric sensor. The step-by-step procedure of the measurement strategy is the following:
**Step** **1**:The initial blood pressure is measured by the cuff method.**Step** **2**:The pressure signals from the radial artery are converted to an electrical signal through the piezoelectric sensor. Continuous PPW signals are filtered and amplified by the front-end analog circuit **Step** **3**:The voltage change of SBP and DBP feature points is identified within the adjacent PPW signals (Figure 1a).**Step** **4**:The voltage change (ΔmV) of SBP and DBP feature points are converted to pressure change (ΔmmHg) of SBP and DBP by the pressure sensitivity of the piezoelectric sensor (Figure 1b).**Step** **5**:Sum of initial blood pressure by cuff method and pressure changes by the piezoelectric sensor to achieve beat-to-beat SBP and DBP monitoring, according to Equations (2) and (3) (Figure 2)
(2)$${SBP(t)\;=\;SBP}_{\mathrm{cuff},\;\mathrm{initial}}+\;\Delta {\mathrm{mmHg}}_{1}+\;\Delta {\mathrm{mmHg}}_{2}\;+\ldots$$
(3)$${\mathrm{DBP}(t)\;=\;\mathrm{DBP}}_{\mathrm{cuff},\;\mathrm{initial}}+\;\Delta {\mathrm{mmHg}}_{a}+\;\Delta {\mathrm{mmHg}}_{b}\;+\ldots$$

## 3. System Design

The system can be described in three main parts, including piezoelectric sensor, front-end analog circuit, and software processing unit (Figure 3). The piezoelectric sensor provides a sensing function of pulsation changes of the radial artery that converts pressure signals into electrical signals by the pressure sensitivity of the piezoelectric sensor. The front-end analog circuit is responsible for PPW amplification and filtering. The round piezoelectric sensor and the analog frontend circuit were constructed with geometric dimensions of 2.4 cm in diameter and 5 cm × 7 cm, respectively. The post-processing unit is responsible for detecting SBP and DBP feature points and calculating pressure changes between the adjacent waves.

### 3.1. Piezoelectric Sensor 

The pulsation signal from the radial artery is a weak signal, which is highly susceptible to interference from electric and magnetic fields during the detection procedure. Therefore, the sensitivity, resolution, and anti-interference ability of the sensor must be considered when detecting the PPW signals. A piezoelectric module HK-2000B (Hefei Huake Electronic Technology Research Institute, Hefei, China) was used in this study. The HK-2000B is a medical pulse sensor that uses piezoelectric-sensitive materials of polyvinylidene fluoride (PVDF) film, which directly converts the arterial pressure signals into electrical signals. The main characteristics of HK-2000B are high sensitivity (2 mV/mmHg) and wide pressure detection range (−50–3000 mmHg). The piezoelectric sensor was mounted on the subject’s wrist above the radial artery by a wrist strap, which can maintain stable mechanical coupling of the sensor with the skin (Figure 4).

### 3.2. Front-End Analog Circuit

In order to extract clear PPW signals, a front-end analog circuit is required to process signals obtained from the piezoelectric sensor. The front-end analog circuit can be divided into two parts, including the alternating current (AC)-coupling circuit and amplifier circuit. The AC-coupling circuit is responsible for direct current (DC) removal in bio-potential measurements [19] that provides a differential high-pass filter without the grounded resistors to eliminate baseline wandering in front of the amplifier circuit [20]. The AC-coupling circuit was designed with a cutoff frequency of 0.03 Hz, according to Equation (4). In addition, the high impedance voltage followers were behind the AC-coupling circuit to avoid the signals to attenuate.
(4)$$f_{c}=\;\frac{1}{{2\pi R}_{y}C_{x}}$$

AD620 instrumentation amplifiers (Analog Devices Inc., Norwood, MA, USA) were used for signal amplification with optimal power supply of ±15 V that was provided by the data acquisition (DAQ) device with ±15 V DC power source. The AD620 provides a tunable gain of 1 to 10,000 by an external resistor, according to Equation (5). This study chose the gain of 989 using RG of 50 Ω with a high common-mode rejection ratio (CMRR) of 140 dB. In order to cover PPW signals bandwidth, a combination of a Butterworth high-pass filter (HPF) of 0.05 Hz and low pass filter (LPF) of 35 Hz were implemented, according to Equations (6) and (7). Based on the analysis of hardware functionality, sweeping frequencies from 10^−3^ Hz to 10^3^ Hz were performed to evaluate the frequency response of the front-end analog circuit. The simulation result performed the required bandwidth (0.05–35 Hz) with an amplification factor of about 59.9 dB and obtained a lower cutoff frequency (*f_L_*) of 0.05 Hz and a higher cutoff frequency (*f_H_*) of 35.12 Hz (Figure 5).
(5)$$\mathrm{Gain}=\;\frac{49.4K\Omega }{R_{G}}+1$$
(6)$$f_{\mathrm{HPF}}=\;\frac{1}{2\pi \sqrt{R_{1}R_{2}C_{1}C_{2}}}$$
(7)$$f_{\mathrm{LPF}}=\;\frac{1}{2\pi \sqrt{R_{3}R_{4}C_{3}C_{4}}}$$

### 3.3. Post-Processing Unit 

A custom-designed software based on the LabVIEW platform was developed to analyze the PPW signal in the post-processing unit. To extract the signals of the arterial pressure pulse, the respiration signal removal was required to eliminate the baseline wandering by detrending the raw recording. The signal denoise was important for peak and valley detection algorithm; therefore, the WA Denoise VI function was implemented after the baseline wandering removal. In order to extract the peak and valley points in PPW signals, a threshold detection method was provided (Figure 6). The voltage information in SBP and DBP feature points was described (the solid lines are de-noise PPW signals, and dashed lines are detection thresholds). The voltage change of SBP and DBP feature points within adjacent PPW signals can be acquired, according to Equations (8) and (9). The pressure sensitivity of the piezoelectric sensor offers a unit conversion of 2 mmV/mmHg that implies that the pressure change is 0.5 mmHg per mV, according to Equation (10). Therefore, the pressure change of SBP and DBP feature points between the adjacent PPW signals were obtained by (11).
(8)$$\Delta {mV}_{1}{=V}_{\max,2}{-V}_{\max,1}\;,\;\Delta {mV}_{2}{=V}_{\max,3}{-V}_{\max,2}\;,\;\ldots$$
(9)$$\Delta {mV}_{a}{=V}_{\min,2}{-V}_{\min,1}\;,\;\Delta {mV}_{b}{=V}_{\min,3}{-V}_{\min,2}\;,\;\ldots$$
(10)$$\mathrm{Pressure}\;\mathrm{sensitivity}\;=\;\frac{2\mathrm{mV}}{\mathrm{mmHg}}$$
(11)$$\Delta \mathrm{mmHg}\;=\;\frac{\Delta V\;\times \;1000}{\mathrm{Gain}}{\;\times (\mathrm{Pressure}\;\mathrm{sensitivity})}^{-1}$$

### 3.4. Ethics Statement

The experiment was approved by the Institutional Review Board of National Chiao Tung University (registration number: NCTU-REC-108-087E). Thirty subjects participated in the experiment (19 males and 11 females, age 20–60 years, height 150–183 cm, weight 50–110 kg). They were healthy without any known diseases and provided their written informed consent. They were asked to refrain from alcohol, caffeine, and strenuous exercise for 1 h before the measurement. In the experiment, all subjects consented to participate and were instructed to remain in a sitting position without movement to avoid motion artifacts during the measurement.

## 4. Experimental results

### 4.1. Pressure Pulse Wave (PPW) Signals Analysis for Beat-to-Beat Systolic Blood Pressure (SBP) and Diastolic Blood Pressure (DBP)

The piezoelectric sensor was placed on the skin above the radial artery and measured for 10 sec to obtain the continuous pressure waveform. Figure 7a shows the continuous PPW signals obtained from the analog circuit. In order to detect the exact locations of the peak and valley of PPW signals, post-processing was performed to produce distinguishable feature points of SBP and DBP, as shown in Figure 7b. Figure 8 shows that the peak and valley values were extracted from the continuous PPW post-processing signals by the threshold method. Figure 9a indicates that the voltage change within adjacent PPWs was calculated, according to Equations (8) and (9). The pressure change within adjacent PPWs was obtained by conversion of pressure sensitivity (2 mV/mmHg) and circuit amplification gain of 989, according to Equation (11). Therefore, the pressure change between adjacent beats can be calculated, as shown in Figure 9b. The measurement strategy uses the sum of the initial blood pressure by an oscillometric method and pressure change by a piezoelectric sensor to achieve the beat-to-beat SBP and DBP monitoring, according to Equation (1). The initial SBP and DBP measured by the cuff-method were respectively 109 mmHg and 61 mmHg. Figure 10 demonstrates the beat-to-beat SBP and DBP monitoring.

### 4.2. Continuous Blood Pressure Measurement and Accuracy Evaluation 

In order to perform accuracy evaluation compared to the oscillometric method, the left and right hands of 30 subjects with the individual sensors were synchronously measured for 30 min. The piezoelectric sensor was placed on the left hand to measure the continuously beat-to-beat pressure change for 30 min. The oscillometric sensor was measured on the right hand at a regular interval of 1 min for 30 measurements in 30 min (Figure 11). The performance of our system was evaluated by recording the MAE and standard deviation (SD) within two methods, according to Equations (12) and (13).
(12)$$\mathrm{MAE}=\frac{1}{n}\overset{n}{\underset{i=1}{\sum }}{|y}_{i}{-x}_{i}|\;$$
(13)$$\mathrm{SD}=\sqrt{\frac{\sum_{i=1}^{n}{{(y}_{i}{-x}_{i}-\mathrm{MAE})}^{2}}{n-1}}$$
where *xi* is the reference blood pressure values obtained from the oscillometric sensor and *yi* is the measurement blood pressure values obtained from our system on *n* measurements. Table 1 demonstrates the average MAE ± SD of SBP and DBP were 1.52 ± 0.30 mmHg and 1.83 ± 0.50 mmHg for 30 subjects.

## 5. Discussions 

### 5.1. Measurement Strategy and Performance 

This study developed a wearable piezoelectric-based system that uses the sum of the initial blood pressure by an oscillometric method and pressure change by a piezoelectric sensor to obtain the beat-to-beat SBP and DBP. The continuous PPW signals extracted from the radial artery by a piezoelectric sensor directly reveals the arterial behaviors of expansion and contraction. The beat-to-beat pressure change was obtained by the pressure sensitivity of the piezoelectric sensor (2 mV/mmHg). In order to linearly convert the electrical signals to pressure signals by pressure sensitivity, the tested signals must stay within the pressure range of the sensor in the specification. The ranges of SBP and DBP are, respectively, from 70/40 mmHg and 190/100 mmHg, including categories of hypotension, normal blood pressure, and hypertension. We used a piezoelectric sensor with a pressure range of −50–3000 mmHg to ensure the system operation in the linear region; therefore, we believe that our method can obtain correct blood pressure values in different cardiovascular diseases by the combined evidence of sensor characteristics and experimental results. The experimental results indicated that the reliable MAE ± SD for the SBP and DBP were 1.52 ± 0.30 mmHg and 1.83 ± 0.50 mmHg, which agreed with the Association for the Advancement of Medical Instrumentation (AAMI) [21] and British Hypertension Society (BHS) [22] standard criteria of MAE within 5 mmHg and SD within ±8 mmHg. Therefore, this study demonstrates a novel measurement strategy that uses sensor properties to obtain accurate beat-to-beat blood pressure recording without multiple calibrations and complex regression analysis of the estimation model.

### 5.2. Accuracy Evaluation with Other Works 

A summary of recent studies on continuous blood pressure measurement is shown in Table 2. These works demonstrated the proposed technique with accuracy compared to cuff-sensor. Lazazzera et al. [9] presented a smartwatch of CareUp® (Farasha Labs, Paris, France) for blood pressure estimation that uses the ECG and PPG physiological channels to calculate the PTT and estimate the blood pressure by a linear model. The experimental results indicated that the MAE ± SD for SBP and DBP were validated by a sphygmomanometer, and obtained 1.52 ± 9.45 mmHg and 0.39 ± 4.93 mmHg on 44 subjects. Slapniˇcar et al. [10] created a blood pressure estimation model by PPG measurement and deep neural network. The results obtained MAE for SBP and DBP were, respectively, 9.43 and 6.88. Simjanoska et al. [23] provided a machine learning model to estimate the blood pressure from ECG recording on 51 different subjects and obtained MAE ± SD for SBP and DBP were 7.72 ± 10.22 mmHg and 9.45 ± 10.03 mmHg. Wang et al. [24] used PPW and PPG wave to calculate the PTT and established the blood pressure estimation model and resulted in an accuracy of 3.71 ± 3.06 mmHg for SBP and 5.44 ± 5.10 mmHg for DBP. Liu et al. [25] used impedance plethysmography (IPG) technique to establish SBP model by linear regression and obtained the correlation coefficient of 0.7 with a cuff method. Liu et al. [17] provided a multiparameter fusion (MPF) estimation model that combined 21 features of PPW and computed the summation of 21 features of PPW for blood pressure estimation. The PPW-based mathematical model was used for further comparative analysis with the best PTT-based model and obtained the excellent estimation error of 0.7 ± 7.78 mmHg for SBP, and 0.83 ± 5.45 mmHg for DBP, compared to the PTT-based model of 1.33 ± 0.37 mmHg and 1.14 ± 0.20 mmHg for SBP and DBP.

Compared to the above studies, we demonstrated a low-cost and accurate continuous beat-to-beat SBP/DBP detection system requiring fewer PPW features. Importantly, our system could obtain a qualified accuracy of MAE within 2 mmHg and SD within ± 1 mmHg without the need for multiple calibrations and complex regressions. We believe that our method is novel in that it is a step further than the previous studies.

### 5.3. Limitation

We used a straightforward method to convert voltage changes to pressure changes in PPW signals directly by piezoelectric properties of pressure sensitivity. However, stable PPW signals from the radial artery remain a significant issue that can also be affected by daily activities such as muscle changes caused by walking, eating, dressing, grabbing, etc. Our study used a solid piezoelectric sensor with an analog front-end to validate the direct estimation concept with the scope of measurement in the resting condition lasting for up to 30 min. No doubt, the solid structure of the piezoelectric sensor could easily induce air gaps under extended use during daily activities. In our future work, we expect to (1) improve the design of flexible piezoelectric sensors, and optimized analog frontend construction may more appropriately address this important aspect in ambulatory blood pressure measurement, and (2) recruit more participants, especially patients with blood pressure problems, to make our method more reliable.

## 6. Conclusions

This paper develops a low-cost piezoelectric-based system to validate a straightforward computation for continuous beat-to-beat blood pressure measurement. The method uses the summation of initial blood pressure and pressure change without the need for calculating correlation and repeated calibrations. We estimated the continuous beat-to-beat blood pressure using the pressure sensitivity of the piezoelectric sensor to directly convert the voltage difference between adjacent systolic and diastolic feature points in PPW into pressure difference. The experimental results indicated our system produced a reliable accuracy of SBP (MAE ± SD 1.52 ± 0.30 mmHg) and DBP (MAE ± SD 1.83 ± 0.50 mmHg), which agreed with the AAMI of MAE within 5 mmHg and SD within ± 8 mmHg. Overall, this study developed a low-cost wearable piezoelectric-based system using an intuitive measurement strategy with qualified accuracy. The new system is potentially suitable for continuous long-term blood pressure-monitoring applications.

## Figures and Tables

**Figure 1 sensors-20-00851-f001:**
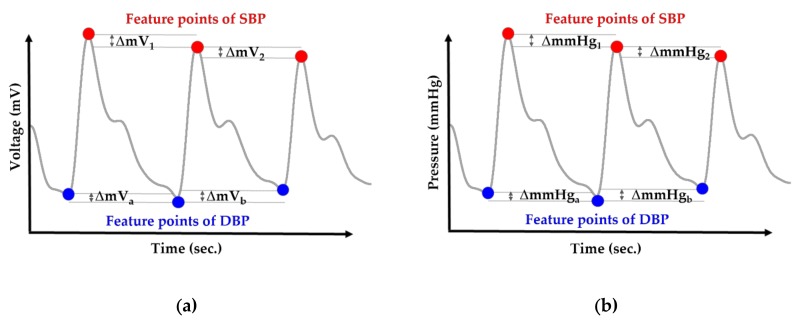
(**a**) Calculate the ΔmV at the systolic blood pressure (SBP) and diastolic blood pressure (DBP) feature points within the adjacent wave. (**b**) Convert the voltage change (ΔmV) into pressure change (ΔmmHg) by the pressure sensitivity of the piezoelectric sensor (mV/mmHg).

**Figure 2 sensors-20-00851-f002:**
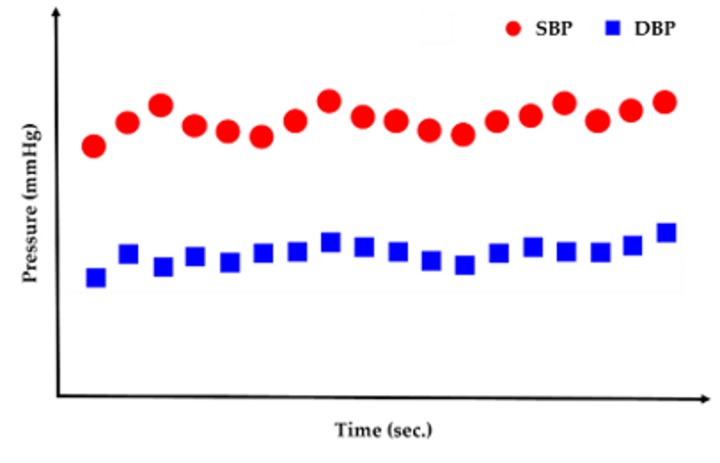
Continuous beat-to-beat SBP and DBP monitoring.

**Figure 3 sensors-20-00851-f003:**
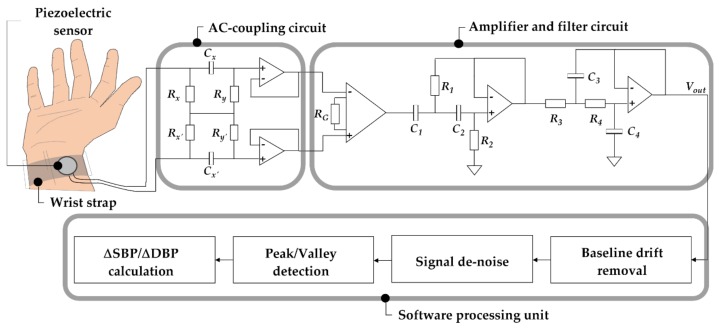
Schematic of the overall system.

**Figure 4 sensors-20-00851-f004:**
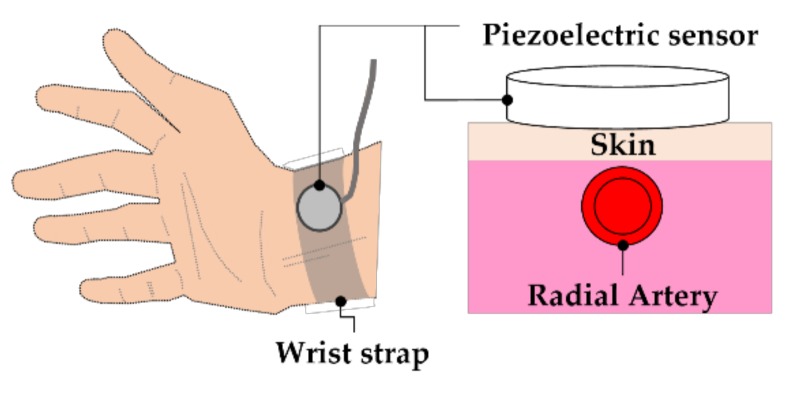
Piezoelectric sensor measurement location.

**Figure 5 sensors-20-00851-f005:**
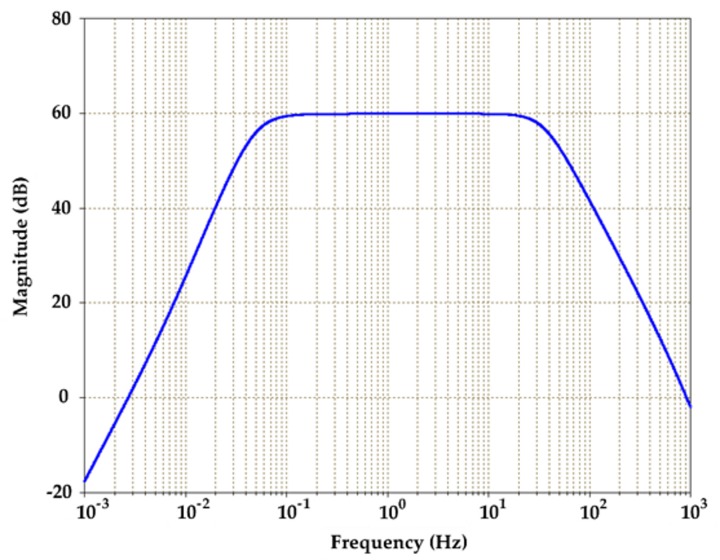
The simulated frequency response of the system.

**Figure 6 sensors-20-00851-f006:**
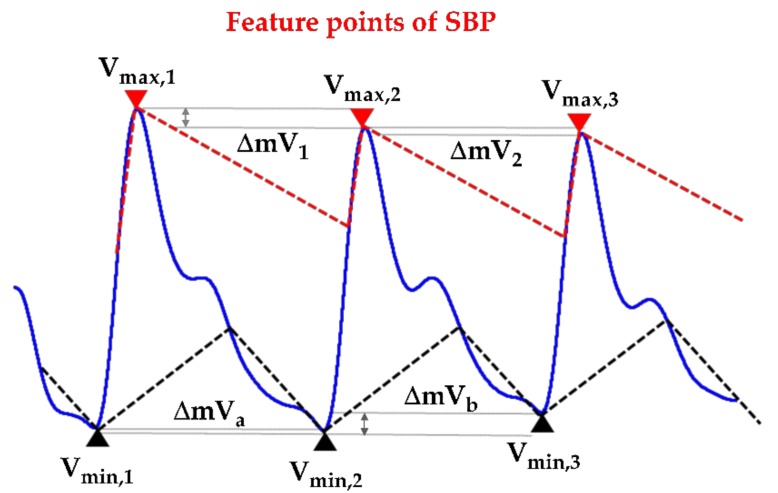
Peak and valley detection algorithm for pressure pulse wave (PPW) signals.

**Figure 7 sensors-20-00851-f007:**
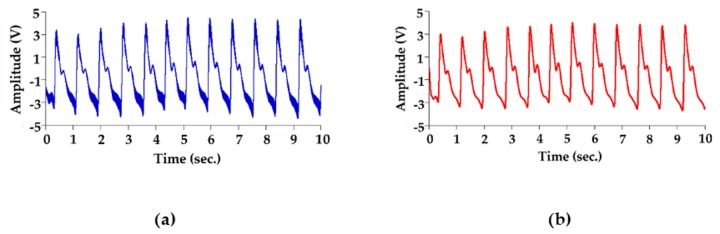
PPW signals from the (**a**) analog circuit (**b**) post-processing unit.

**Figure 8 sensors-20-00851-f008:**
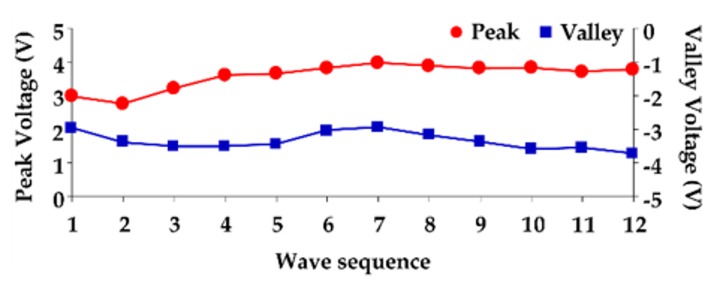
Peak and valley detection algorithm for PPW signals.

**Figure 9 sensors-20-00851-f009:**
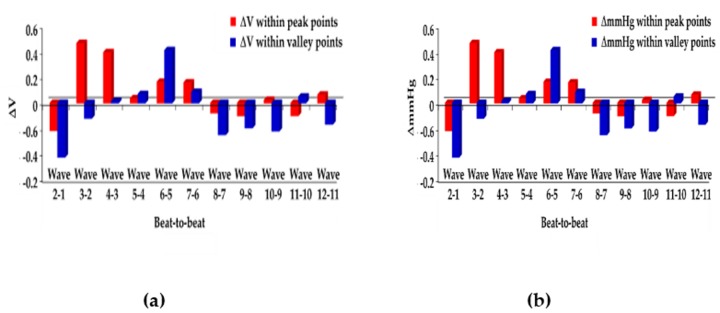
(**a**) The voltage change and (**b**) pressure change within feature points of SBP and DBP.

**Figure 10 sensors-20-00851-f010:**
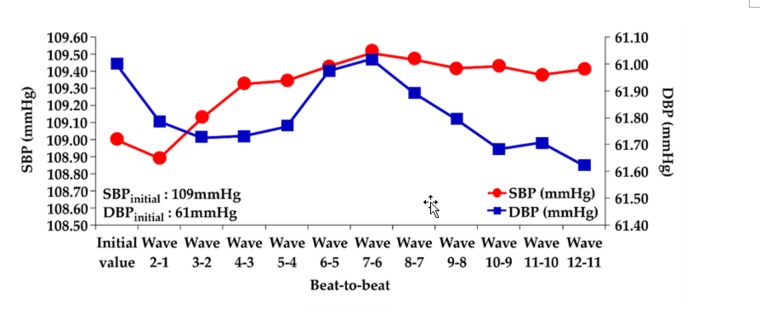
Beat-to beat blood pressure for 12 beats.

**Figure 11 sensors-20-00851-f011:**
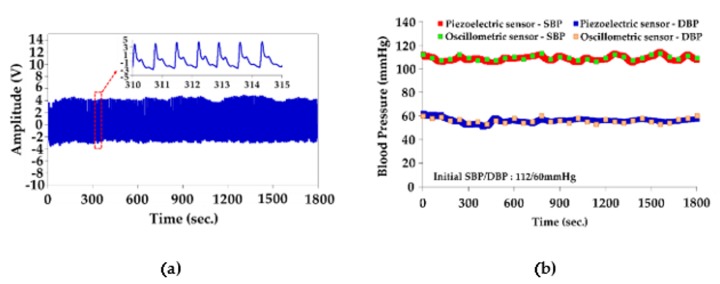
(**a**) PPW signals from the radial artery for 30 min measurement. (**b**) Accuracy evaluation for a piezoelectric-base system for 30 min measurement, compared to an oscillometric sensor (Subject 1).

**Table 1 sensors-20-00851-t001:** Accuracy evaluation for our system.

Subject	Male	Age	Height (cm)	Weight (kg)	SBP Error (MAE ± SD mmHg)	DBP Error (MAE ± SD mmHg)
1	Male	58	180	90	1.34 ± 0.07	1.61 ± 0.16
2	Female	57	150	68	1.51 ± 0.13	1.92 ± 0.22
3	Male	60	170	80	1.46 ± 0.17	1.87 ± 0.29
4	Female	36	155	66	1.62 ± 0.21	2.01 ± 0.38
5	Male	31	168	65	1.24 ± 0.27	1.56 ± 0.39
6	Male	36	175	90	1.44 ± 0.33	1.64 ± 0.66
7	Male	27	175	68	1.47 ± 0.22	1.77 ± 0.52
8	Female	27	156	55	1.7 ± 0.42	2.06 ± 0.71
9	Female	24	160	90	1.39 ± 0.35	1.72 ± 0.66
10	Male	28	180	110	1.58 ± 0.44	1.95 ± 0.77
11	Male	36	176	105	1.22 ± 0.21	1.84 ± 0.51
12	Female	27	163	55	1.38 ± 0.19	1.79 ± 0.44
13	Female	27	164	61	1.66 ± 0.47	1.68 ± 0.41
14	Male	27	168	68	1.53 ± 0.38	1.72 ± 0.49
15	Male	27	165	79	1.47 ± 0.24	1.76 ± 0.39
16	Male	27	168	72	1.51 ± 0.26	1.89 ± 0.44
17	Male	27	170	68	1.42 ± 0.33	1.77 ± 0.46
18	Female	59	156	71	1.45 ± 0.22	1.87 ± 0.34
19	Female	54	154	50	1.38 ± 0.28	1.92 ± 0.44
20	Male	23	168	66	1.54 ± 0.36	1.86 ± 0.55
21	Male	27	163	62	1.58 ± 0.32	1.83 ± 0.51
22	Male	23	178	72	1.56 ± 0.28	1.78 ± 0.61
23	Male	20	173	68	1.33 ± 0.15	1.82 ± 0.53
24	Female	21	160	58	1.49 ± 0.24	1.79 ± 0.46
25	Female	27	160	54	1.62 ± 0.33	1.96 ± 0.61
26	Male	20	175	53	1.84 ± 0.37	1.77 ± 0.55
27	Male	20	174	105	1.58 ± 0.41	2.07 ± 0.66
28	Male	23	183	71	1.77 ± 0.55	1.88 ± 0.73
29	Male	20	170	85	1.67 ± 0.41	1.83 ± 0.55
30	Female	20	155	61	1.89 ± 0.43	1.88 ± 0.62
Average					1.52 ± 0.30	1.83 ± 0.50

**Table 2 sensors-20-00851-t002:** Recent works for blood pressure measurement.

Author	Technique	Statistic Method	SBP Error (mmHg)	DBP Error (mmHg)	Ref.
Lazazzera	${\mathrm{PTT}}_{\mathrm{ECG}.\;\mathrm{PPG}}$	MAE ± SD	1.52 ± 9.45	0.39 ± 4.93	[9]
Simjanoska	${\mathrm{PTT}}_{\mathrm{ECG}}$	MAE ± SD	7.72 ± 10.22	9.45 ± 10.03	[23]
Slapniˇcar	${\mathrm{PTT}}_{\mathrm{PPG}}$	MAE	9.43	6.88	[10]
Wang	${\mathrm{PTT}}_{\mathrm{PPW},\;\mathrm{PPG}}$	MAE ± SD	3.71 ± 3.06	5.44 ± 5.10	[24]
Liu	${\mathrm{PTT}}_{\mathrm{IPG}}$	Correlation coefficient	0.7	-	[25]
Liu	${\mathrm{PPW}}_{\mathrm{Estimation}\;\mathrm{model}}$	MAE ± SD	0.70 ± 7.78	0.83 ± 5.43	[17]
Our work	${\mathrm{PPW}}_{\mathrm{Pressure}\;\mathrm{change}}$	MAE ± SD	1.52 ± 0.30	1.83 ± 0.50	-

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
