# Peer review of "Wearable Piezoelectric-Based System for Continuous Beat-to-Beat Blood Pressure Measurement"

_sensors, 2020, doi:10.3390/s20030851_

Round 1

Reviewer 1 Report

Dear Authors, thank you for your comprehensive response to all my questions and remarks. In my opinion, the quality of the manuscript has been significantly improved (much better-written text; better quality illustrations; all changes suggested by the reviewer have been taken into account by the authors), and it is worth publishing in MDPI Sensors.

Author Response

Answer to Comments

Sensors - Manuscript ID: sensors-709860

Reviewer 1:

Dear Authors, thank you for your comprehensive response to all my questions and remarks. In my opinion, the quality of the manuscript has been significantly improved (much better-written text; better quality illustrations; all changes suggested by the reviewer have been taken into account by the authors), and it is worth publishing in MDPI Sensors.

We thank the reviewer for the careful review.

Reviewer 2 Report

The authors developed a piezoelectric-based system for pressure measurement. The article is well written.

One of the issues regarding the use of piezoelectric transducer is related to the mechanical coupling. Please, provide more details about the mounting method on the skin. How can the authors guarantee that the coupling is will not alter the measurements? Please, provide more technical details about the piezoelectric used in the work (frequency response, etc).

Author Response

Answer to Comments

Sensors - Manuscript ID: sensors-709860

Reviewer 2:

The authors developed a piezoelectric-based system for pressure measurement. The article is well written.

One of the issues regarding the use of piezoelectric transducer is related to the mechanical coupling. Please, provide more details about the mounting method on the skin. How can the authors guarantee that the coupling is will not alter the measurements? Please, provide more technical details about the piezoelectric used in the work (frequency response, etc).

We appreciate the reviewer for the encouragement of our study. The piezoelectric sensor was mounted on the subject’s wrist above the radial artery by a wrist strap, which can maintain stable mechanical coupling of the sensor with skin (Figure 4). We have revised Figure 4 and added a description of the mounting method for the piezoelectric sensor on page 4, line 125-127. Moreover, all subjects were instructed to remain in a sitting position without movement during the blood pressure measurement (page 6, line 175-177). In doing so, we made sure that the coupling condition would not alter the measurements.

We have added Figure 5 to show the simulation results of the frequency response of the front-end analog circuit (page 5, line 144-148).

This manuscript is a resubmission of an earlier submission. The following is a list of the peer review reports and author responses from that submission.

Round 1

Reviewer 1 Report

I noticed that the newly revised manuscript did not improve significantly compared to the original manuscript, and there were still serious flaws and incomplete experiments. The paper should not be considered for publication in Sensors.

Author Response

Reviewer: 1

Point:

I noticed that the newly revised manuscript did not improve significantly compared to the original manuscript, and there were still serious flaws and incomplete experiments. The paper should not be considered for publication in Sensors.

Response:

  We thank the reviewer for the comments on our manuscript. We provided answers to reviewers 2’s comments to address our novelty. Compared to related works, we demonstrated a low-cost and accurate continuous beat-to-beat SBP/DBP detection system with fewer PPW features. Importantly, our system could obtain a qualified accuracy of MAE within 2mmHg and SD within ± 1mmHg without multiple calibrations and complex regressions.

   Even though this study validated the feasibility of the piezoelectric-based system in continuous blood pressure measurement through an intuitive computation on PPW signals, some limitations require further investigation to ensure that the approach is fully applicable in realistic situations. Therefore, we added the description of the limitations of our study on page 9, line 247-257.

Reviewer 2 Report

The article presents an alternative, low-cost method for continuous beat-to-beat blood pressure measurement. The manuscript presents a good scientific level; its structure is correct and logical. The methods used and the results obtained are clearly described.

Remarks and comments:
================
1) Please add information on the sensitivity of the piezoelectric sensor used.

2) Do hand movements, clenching fists, etc. causes excitation of the piezoelectric sensor?

3) Good mechanical coupling of the sensor with the skin is ensured by the wrist strap. How is acoustic coupling implemented? The issue is particularly important in view of the long-term use of the system. Is an ordinary ultrasound gel enough?

4) Figure 4(b): The schematic diagram of how the piezoelectric sensor works (piezoelectric effect) is somewhat trivial and probably unnecessary.

5) The optimal power supply for the AD620 measurement amplifier is ±15 V, which seems quite problematic. How was the power supply for the signal conditioning module implemented?? Where is the power module located/mounted?

6) Figure 7: I suggest showing the amplitude values using two Y axes - separately for peaks and valleys. The characteristics will be less "flat" and easier to analyze.

7) Figure 7 and 8: "valleys" instead of "volleys"

8) In the text, I did not find information about the method of noise reduction used. I suspect that the wavelet method (DWT or UWT) was chosen, which is available in Labview (WA Denoise VI) ...?

Author Response

Reviewer: 2

The article presents an alternative, low-cost method for continuous beat-to-beat blood pressure measurement. The manuscript presents a good scientific level; its structure is correct and logical. The methods used and the results obtained are clearly described.

        We thank the reviewer for the encouragement of our work. The point-by-point answers to comments are shown below.

Remarks and comments:

================

1) Please add information on the sensitivity of the piezoelectric sensor used.

        We have addressed the important characteristics of the pressure sensitivity of the piezoelectric sensor on page 4, line 126-127. Importantly, we also indicated the applicable region in pressure sensitivity and ensured the piezoelectric sensor to operate with the pressure sensitivity of 2mV/mmHg in the linear region with a wider pressure range of -50 to 3000mmHg that sufficiently satisfied different clinical situations, including categories of hypotension, normal blood pressure, and hypertension. We added the description of the pressure sensitivity of the piezoelectric sensor on page 8, line 210-215.

2) Do hand movements, clenching fists, etc. causes excitation of the piezoelectric sensor?

        We fully agree with such an insightful assessment. Stable PPW signals from the radial artery remains a significant issue that can also be highly affected by daily activities such as muscle changes caused by walking, eating, dressing, grapping, etc. Our study used a solid piezoelectric sensor with an analog front-end to validate the direct estimation concept with the scope of measurement in the resting condition lasting for up to 30 minutes. No doubt, the solid structure of the piezoelectric sensor could easily induce air gaps under extended use during daily activities. In our future work, improved design of flexible piezoelectric sensors and optimized analog frontend construction may more appropriately address this important aspect in ambulatory blood pressure measurement. We added the description of the limitations of the ambulatory blood pressure measurement on page 9, line 247-257.

3) Good mechanical coupling of the sensor with the skin is ensured by the wrist strap. How is acoustic coupling implemented? The issue is particularly important in view of the long-term use of the system. Is an ordinary ultrasound gel enough?

        We agree that good mechanical coupling of the sensor with the skin is ensured by the wrist strap. In our study, we use the positive piezoelectric effect of the sensor to extract the voltage through arterial pulsation. We believe that good contact between sensor and skin is a significant condition for long-term blood pressure measurement that directly affects the piezoelectric sensor sensing capability for pulsation signal from the radial artery. Therefore, we use a wrist strap with adjustable tightness to fit the different sizes of users' wrists in the experiments. Nevertheless, a layer of gel may be needed between the transducer and the body to get good acoustic coupling and remove extra air gaps. In our future work, improved design of flexible piezoelectric sensors and optimized analog frontend construction may more appropriately address this important aspect in ambulatory blood pressure measurement.

4) Figure 4(b): The schematic diagram of how the piezoelectric sensor works (piezoelectric effect) is somewhat trivial and probably unnecessary.

        We thank the reviewer for the excellent suggestion. We have removed Figure 4(b) and added a description of the piezoelectric effect on page 4, line 124-126.

5) The optimal power supply for the AD620 measurement amplifier is ±15 V, which seems quite problematic. How was the power supply for the signal conditioning module implemented?? Where is the power module located/mounted?

        AD620 instrumentation amplifiers were used for signal amplification with an optimal power supply of +/-15V that was provided by the data acquisition (DAQ) device with +/- 15V DC power supply. Our current design used external dual power supply to validate the system performance for feasibility study of continuous blood pressure measurement. For future studies, we will consider using single power with low voltage supply to implement our system by optimized circuit design. We added the description of detailed implementation on page 5, line 140-142.

6) Figure 7: I suggest showing the amplitude values using two Y axes - separately for peaks and valleys. The characteristics will be less "flat" and easier to analyze.

        We thank the reviewer for the excellent suggestion. We gratefully follow your advice and revise the figure per your kind advice.

7) Figure 7 and 8: "valleys" instead of "volleys"

        We thank the reviewer for the careful review. We have revised the wording in Figures 7 and 8.

8) In the text, I did not find information about the method of noise reduction used. I suspect that the wavelet method (DWT or UWT) was chosen, which is available in Labview (WA Denoise VI) ...?     

        We have described the noise reduction method of WA Denoise VI on page 5, line 153-154.
